# Identifying Key Nodes for the Influence Spread Using a Machine Learning Approach

**DOI:** 10.3390/e26110955

**Published:** 2024-11-06

**Authors:** Mateusz Stolarski, Adam Piróg, Piotr Bródka

**Affiliations:** 1Department of Artificial Intelligence, Wrocław University of Science and Technology, 50-370 Wrocław, Poland; piotr.brodka@pwr.edu.pl; 24Semantics, 00-833 Warszawa, Poland; a.pirog@4semantics.com

**Keywords:** social networks, node classification, unsupervised learning, influence spread

## Abstract

The identification of key nodes in complex networks is an important topic in many network science areas. It is vital to a variety of real-world applications, including viral marketing, epidemic spreading and influence maximization. In recent years, machine learning algorithms have proven to outperform the conventional, centrality-based methods in accuracy and consistency, but this approach still requires further refinement. What information about the influencers can be extracted from the network? How can we precisely obtain the labels required for training? Can these models generalize well? In this paper, we answer these questions by presenting an enhanced machine learning-based framework for the influence spread problem. We focus on identifying key nodes for the Independent Cascade model, which is a popular reference method. Our main contribution is an improved process of obtaining the labels required for training by introducing “Smart Bins” and proving their advantage over known methods. Next, we show that our methodology allows ML models to not only predict the influence of a given node, but to also determine other characteristics of the spreading process—which is another novelty to the relevant literature. Finally, we extensively test our framework and its ability to generalize beyond complex networks of different types and sizes, gaining important insight into the properties of these methods.

## 1. Introduction

The significant growth of online societies has created new directions of research in the field of network science. In recent years, the task of classifying influential nodes has gained popularity as a result of the emerging need to quickly identify key nodes in the spreading process [1]. The purpose of key node identification, also known as seed selection [2], is to find nodes that can maximize the reach (i.e., the number of affected nodes) and/or the velocity of the spreading process within a given budget—the number of key nodes we can afford to activate. Influence spread is also well known in the field of information entropy, introduced by Claude Shannon [3], which states that the emergence of a new event from a communicated message is influenced by how surprising the message’s content is [4]. This task has many practical applications, such as in effective marketing strategies based on cooperation with popular individuals, or even in preventing the spread of an infectious disease. With the rapid expansion of virtual worlds and progressive globalization, the number of real-life use cases for this solution just keeps increasing.

Traditionally, the best method to estimate a given node influence was simply to simulate the spread started at that node [5]. Unfortunately, while easy to implement, this Monte Carlo-type approach had every flaw from this family of methods. It took a tremendous amount of time and computational power to acquire reasonable approximation; still, the obtained results were bound to a specific scenario, with no way of generalizing extracted knowledge beyond this specific network. The alternative approach relies on obtaining the influence evaluation directly from the centrality measures of the network nodes. While drastically reducing the time and computational power, this approach produces significantly less accurate results. Additionally, heavy reliance on chosen centrality measures can make them overly sensitive to specific cases [6,7]. The modern approach of applying machine learning algorithms proved to be capable of outperforming previous methods in accuracy and consistency [8,9,10,11,12,13,14]. However, certain limitations exist in the process of obtaining labels, which are reflected in the inaccurate and overly simplified identification of the key nodes [8,10]. In addition, ML methods have the ability to generalize and reuse previously obtained knowledge—this allows us to evaluate previously unseen nodes, or even whole networks, in a very short time (without the need for rerunning expensive simulations), while maintaining high accuracy. Although machine learning introduced significant advancements to this field, further research is required before they can be easily utilized in everyday applications.

This study proposes an enhanced machine learning-based framework for influential node identification and classification, available at https://github.com/mateuszStolarski/identifying-key-nodes-influence-spread-ml (accessed on 24 October 2024). We analysed currently used approaches, pinpointed their weaknesses and proposed better alternatives. The experiments we performed on real graphs showed significant improvement using the proposed features.

To summarize, our main contributions to the field are as follows:We introduce a novel node labelling approach based on unsupervised machine learning, Smart Bins—a significant improvement over techniques used in the modern relevant literature.We propose alternative ways to define influential nodes by selecting them on the basis of spreading peak and time, which have not yet been used.We examine the proposed framework’s ability to train a model that can generalize beyond the network used for its training.We conduct a broad feature importance analysis, which allows us to select a group of the most crucial centrality methods for key node identification and classification in further research.

Together, these elements form an improved approach for detecting influential nodes, which is able to achieve higher and more stable results than currently existing works.

The rest of this paper is organized as follows: The next section presents the related works in this area. Section 3 presents our key node detection framework in a step-by-step manner while outlining the differences to the currently used approaches. Section 4 is dedicated to the experiments—a description of the experimental setup, results and analysis. Finally, in Section 5, we conclude our findings and present possible directions for further work.

## 2. Related Work

Numerous studies have proposed various approaches for seed node selection. These approaches can be classified into four basic categories: microstructure-based (MSB), community structure-based (CSB), macrostructure-based (MASB) and machine learning-based (MLB) [6]. MSB algorithms are designed to optimize efficiency in analysing large-scale networks. In order to achieve this, some MSB methods sacrifice a broader perspective in favour of reduced complexity. One example of a simple MSB method is degree centrality, as described by Freeman [15]. In exchange for low complexity, such methods have more challenges in terms of high accuracy. In addition, information entropy has also been used to evaluate the spreading potential of nodes in complex networks by determining the node’s influence using centralities based on their two-hop neighbours [16,17]. Unlike the MSB approach, both CSB and MASB algorithms struggle with high complexity, but their use of micro (community) or macro information about network topology can give them advantages in case-specific scenarios, for example, in densely connected networks [18] or in detecting key nodes that typical centrality measures may ignore [6]. Recently, the last category of those algorithms, i.e., machine learning-based, has received a lot of attention because of their ability to make better predictions than MSB algorithms and better generalization than CSB or MASB approaches [8,10], while reducing the computational costs by efficiently reusing previously obtained knowledge. This is because they can discover more topological information than previous methods [19]. Presently, individual centrality methods are no longer competitive with machine learning models due to their inconsistent predictive capabilities across networks [8].

There have been several attempts to apply deep learning, including graph neural networks, to the task of identifying influential nodes, yet most of them have suffered from certain drawbacks. An example of such would be the use of only single centrality in node embedding, which limits their potential compared to the integration of network science methods in deep learning models [20,21]. Moreover, current approaches for obtaining the labels are inadequate due to their reliance on simplistic discretization methods [8,9,10] in the dataset preparation phase. In this paper, we will address these drawbacks and present better alternatives.

## 3. Framework Description

This work proposes an advanced machine learning-based framework for identifying influential nodes in complex networks, as illustrated in Figure 1. This section is dedicated to a detailed introduction of every step and an explanation of our motivations for each step. First, we will discuss the estimation of node influence, including the diffusion model used, the rationale for its selection, and our classification tasks. Next, we describe the label acquisition process in detail, emphasizing how our approach outperforms current methods for influential node identification. Finally, we will present the centrality features we selected for training machine learning models and the preprocessing step. These steps comprise a pipeline capable of processing any network to prepare a dataset suitable for training machine learning models.

### 3.1. Estimating Nodes’ Influence

The approximation of the node power to spread influence in the network is the most important part of network preparation for key node classification. There are several diffusion models for which the choice can be crucial for the reliable mapping of network behaviour. The most commonly used model in related works is the SIR model [6]. This model consists of three states, Susceptible, Infected and Recovered (also known as Removed), and two parameters, infection rate β, which describes the probability of a susceptible node to become infected, and recovery rate γ, which describes the probability that an infected node would recover. The iteration of this diffusion model can be expressed by the following equation:$$\frac{ds}{dt}=-\beta is,\frac{di}{dt}=-\beta is-\gamma i,\frac{dr}{dt}=\gamma i,$$
where *s*, *i* and *r* refer to the states of a node that the node can take during the simulation. The process runs until there are no more infected nodes (or a predefined number of iterations).

*The Independent Cascade* (IC) model has a similar behaviour to the SIR model with a recovery rate equal to 1.0 since after activation (equivalent to infection in the SIR model), each node has only one chance (iteration) to affect/activate its neighbours, and after that iteration, it cannot affect anyone else (equivalent to recovery in the SIR model) [5].

Due to certain characteristics, it is more suited for influence spread modelling than the SIR model. The IC model allows for simpler assumptions when generating labels in our work. By dropping an additional parameter γ (recovery rate) used in the SIR model, we could focus more on the differences between spreading processes in different network types. There are clear contrasts between citation and social networks, as presented in Table 1,which is why using unified sets of activation probabilities would not be appropriate. Moreover, in the reviewed literature, all of the previous studies used the SIR model, wherein the recovery rate was set to 1.0, i.e., after only one iteration of propagation, an infected node would stop spreading and change its state to recovered [8,10,22,23,24], effectively reducing the SIR model to the IC model. Thus, due to the reasons mentioned above, in our experiments, we decided to use the Independent Cascade model. However, our framework does not exclude the usage of other diffusion models, like, for example, the SI model.

To obtain the ground-truth values for a given node, we simulated a spreading process starting at that node and measured its characteristics. We used separate sets of thresholds (activation probabilities) for the tested network types due to their topological differences and interpretations of influence propagation. This set is 0.2, 0.3 and 0.4 for citation networks and 0.1, 0.15 and 0.2 for social networks. We ran the diffusion model 100 times for each node and each threshold, and calculated the means of the raw results to aggregate scores for our key node classification tasks. This enabled us to construct ML models, as described in Section 4.1, with the capability to generalize across previously unseen graphs—thereby optimizing inference time in comparison to non-machine learning methods. By analysing the information obtained from the diffusion model simulations, we propose novel classification tasks that enable the identification of key nodes based on various criteria. These tasks aim to identify nodes that can generate impact on the network as quickly and effectively as possible. The tasks analysed in our experiments include the following:**Influence range:** Predicting the total range, i.e., the number of influenced nodes in the network. This is the main task and is widely recognized in the relevant literature.**Influence peak:** Predicting the maximal number of nodes activated in one iteration. A crucial characteristic from a practical perspective (e.g., the total number of people infected by a virus at once is predicted, which allows us to approximate the workload for medical services).**Peak time:** Predicting the number of iterations required for the process to reach the Influence peak. An important value for real-life use cases.

While the total range generated by the seed is important, it is equally vital to understand how quickly and to what extent we can maximize our influence. This is particularly critical when we aim to maximize the reach of our activities before a specific deadline. According to our state of knowledge, *Influence peak* and *Peak time* have not been previously explored in the key node detection literature.

### 3.2. Obtaining the Labels

The results of the Independent Cascade simulation require further postprocessing before feeding them into an ML model. Formulating the problem as is, i.e., training a regression model to predict the influence range of each node, would cause multiple problems due to the fine granularity of the labels. Additionally, from a practical perspective, on a scale of thousands, an additional influence range of 1 or 2 extra nodes presents no practical difference—it could even be accounted as the margin of error of the Independent Cascade model. Such granularity would also unreasonably increase the learning challenge for the ML algorithm. Instead, a better approach is to discretize the nodes based on their influence range value, thus reformulating the problem as one which is a classification instead of a regression type.

Usually, the approach is to use the nodes’ influence range values to create some number of bins (also known as buckets). In our case, a higher cluster number means a better spreader, e.g., in the first bin, we have the lowest values; in the second, middle values; and in the third, we have the highest values (please note that strictly speaking, each bin is represented by some range, e.g., from 5 to 10, and not the individual values for each node). Based on the division into bins, we can assign labels (classes), which will allow the machine learning algorithm to supervise itself during training and validation (e.g., if a node influence range value fits into the first bucket, it has a label 3, meaning that it is one of the key/most influential nodes and would be a good choice to start the spreading). The main problem is how to divide nodes’ influence range values into these bins so that they are neither too narrow nor too wide, which would result in too many or too few nodes in the last class (including the most influential nodes, i.e., those with label 3).

Bucur [8] approached the problem as a form of a binary classification—after making an assumption that the most influential class (bin) is represented by an arbitrary percentile of the network, she proposed a classifier trying to identify the top 5% of the spreaders. This idea, while easy to implement, lacks the ability to classify other nodes properly. As presented in Table 2, in reality, the distinct group of top spreaders is often much smaller, making the assumption of “5%” far-fetched. What is even more important is that an arbitrary value reduces the flexibility of the method, forcing the models to operate in a fixed setting that may not be reflected in real networks.

Zhao et al. [10] proposed a less cumbersome approach; they suggested using traditional discretization methods, such as uniform or quantile discretization, to obtain a set of *K* classes (*K* is the number of classes/bins). This greatly increased the flexibility of the method, allowing for the identification of all classes of nodes properly but still without addressing the issue of taking the top class as an arbitrary number of points.

To solve this problem, we propose a new approach, *Smart Bins*, which is our main contribution. By utilizing unsupervised machine learning algorithms, we can achieve a flexible discretization based on actual dependencies in the data, not just an arbitrary choice of a parameter. We ran the KMeans algorithm on the results of the spreading model (all node influence ranges) to achieve this effect, with the parameter *k* denoting the number of discretization bins (and in consequence, classes/labels) that we wanted to achieve. Then, we assigned each cluster member a centroid (cluster centre) value.

KMeans is a well-known clustering algorithm [25] that iteratively groups a bigger set of data points into smaller subsets (called clusters) in such a way that points within a single cluster are more similar (according to a given metric) to each other than to representatives of other clusters. The determination of the cluster membership *C* with sample mean μ can be represented by the following formula:$$\sum_{i=0}^{n}\underset{\mu_{j}\in C}{min}\left(|\right|x_{i}-\mu_{j}{\left|\right|}^{2})$$

This model was chosen here for its simplicity and unmatched speed while maintaining reasonably accurate partitioning, but other clustering algorithms can be considered, like Spectral Clustering [26] or DBScan [27].

Using unsupervised learning techniques, we can easily identify distinct groups in the distribution, thus properly drawing a dividing line between classes. An example of such a discretization can be seen in Figure 2, where each class is fitted to a group with a given density. A dashed line marks the top 5% of the nodes, visualizing how cumbersome this arbitrary choice might be. Even in binary discretization, the top class contains only 4.25% of nodes. Beyond theoretical explanation, our experiments empirically confirmed this approach to surpass other methods; the results can be seen in Section 4.3. Utilizing the method of *Smart Binning*, (clustering discretization), we obtained a set of *K* (where *K* denotes the number of bins, i.e., discretization granularity) labels used as ground-truth for the further training of machine learning models. The *K* set is determined based on the number of elements in the bins for the examined *K*. This approach helps to avoid a situation where the granularity is too coarse, resulting in some labels having no elements, which can disrupt the proper training of machine learning models. In our experiments, we limited the maximum *K* to 5 to ensure consistency across all networks. Larger *K* values led to insufficient representation in some cases.

### 3.3. Selecting the Features for Machine Learning Algorithms

The final step to achieve good results with a machine learning model is a trainable representation. We used the centrality measures to create embedding that can describe each node in terms of topology [10]. Among the many existing centrality measures, we selected a set of fourteen that are well known and often used for key node selection. Table 3 presents and briefly describes this set.

Embedding was enhanced with the activation probability used in the diffusion model. This approach allowed correlating the model settings with labels generated for each element in the set. At the end of the data-engineering process, the features were standardized by removing the mean and scaling to unit variance (i.e., applying statistical standardization/Z-score normalization). This was performed in order to facilitate the process of learning and inference of the tested models.

## 4. Experiments

In order to fully test each stage of our influential node identification framework, we conducted a series of experiments evaluating various machine learning algorithms over four real-world networks of characteristics (Table 1):Citeseer [38]—A citation network where edges represent citations between the papers;Pubmed [38]— A citation network where nodes represent publications about diabetes;Facebook [39]—A social network portraying verified pages from the Facebook platform, with likes between them as edges;Github [39]—A social network representing the following between developers, that have at least ten repositories.

All of them have no self loops or parallel and weighted edges for the consistency of comparison. To comprehensively measure the performance of the models, we opted for the *F1 macro* as our chosen evaluation metric, as it is widely recognized for assessing the performance of the classification models.

We tested multiple machine learning algorithms on all these networks, such as logistic regression, K nearest neighbours, Support Vector Classifier, Random Forest Classifier and Gradient Boost Machine (LightGBM) [40]. We deliberately chose these traditional, rather simple, machine learning algorithms in order to focus the attention of the paper on the other steps of the process, which introduce more innovation. The framework itself is flexible enough to be fitted with virtually any given machine learning classifier.

### 4.1. Machine Learning Algorithm Evaluation

Confirming our assumptions, the LightGBM algorithm visibly outperformed other models in all our experiments, as can be seen in Figure 3. This advantage was consistently present in every comparison; therefore, for the sake of clarity, the LightGBM algorithm will be our default ML model in all further experiments.

First, we conducted an experiment based on a classic machine learning scenario. We put aside a random subset of 20% of the graph’s nodes as the hold-out test set, trained the model for each task, and evaluated its performance on the previously unseen part of the data. The results of this experiment, presented in Figure 4, unambiguously show that the model can easily achieve near-perfect performance across all levels of discretization. Furthermore, all the tasks defined by us in Section 3 present a similar level of challenge to the model. The influence range, peak and time needed to achieve this peak can be reliably predicted, thus confirming our thesis and becoming a viable option for real-life use cases.

### 4.2. Model Generalization

Next, we evaluated the model’s ability to generalize beyond a single network. We trained the model on one network and tested its predictive capability on a different network (including networks of different types). The results, seen in Figure 5, provide multiple interesting insights. As expected, this task was visibly more challenging for the ML model. Contrary to the traditional train–test setup, we can observe a notable drop in performance with the increasing level of node discretization. This usually restricts the usability of cross-trained models to 3–4 classes. Nevertheless, the models’ ability to generalize remains significant. As an example, a model trained on *Citeseer* managed to achieve an impressive F1 score of 0.87 when tested on *Facebook*—a network nearly seven times bigger if we look at node count and more than thirty-seven times bigger if we look at edge count.

Analysing the results, we can also see a pattern regarding the type of networks. In each of the experiments, especially considering the higher number of classes, the ML model performed best on the test network from the same family as the train network, i.e., the ML model trained on *Pubmed* had the highest score on *Citeseer* (both being citation networks), the model trained on *Github* performed best on *Facebook* (both being social networks), etc. This pattern has a few deviations at the lower levels of discretization, but this can be explained by the relative simplicity of the task of binary classification; most of the tested models performed well on two classes. This experiment allows us to draw an interesting conclusion that the family and characteristics of the network may be more important than its sheer size—the model trained on *Pubmed* was consistently the best on *Citeseer*, despite *Facebook* being significantly closer regarding network size. This can be a vital observation for practical applications of this method. In real-life scenarios, it may not be necessary to collect enormous amounts of data—instead, it can be better to create a smaller training network with the closest possible similarity to our target network. However, since evaluating network similarity is a complex problem with dozens of different measures [41] to evaluate, we leave out the task of determining which particular network features work the best for generalization for future works.

### 4.3. Impact of Smart Binning

As one of the paper’s main contributions, we proposed the usage of *Smart Binning*—an unsupervised machine learning technique applied to precisely obtain ground-truth values for the further training of ML classifiers. In Section 3.2, we described the idea in detail and outlined the theoretical basis suggesting its superiority over other known methods. As part of the experiments, we conducted a series of tests to empirically evaluate this claim and to compare our approach to that of our competitors. We discretized all our networks using both methods (clustering discretization and arbitrary choice of the top 5% of the nodes like [8,10]) and compared the results of the classification of downstream nodes’ influence.

As seen in Figure 6, clustering discretization visibly outperforms other methods across all test networks. Additionally, *Smart Bins* provide significantly more stable results. This behaviour can be explained by the randomness of arbitrarily chosen bins—during some of the trials, roughly the top 5% happened to be a distinct group of top spreaders, but this could not be guaranteed. Smart Binning, on the other hand, always fits to represent the groups in the best possible way.

### 4.4. The Importance of Features

The final element that we have analysed is the node features’ importance on the ML model output. We used Shapley values [42,43] to compute each feature contribution to the final prediction. They represent how much the presence or absence of feature changes the predicted output of the ML model for a particular input. Shapley values are obtained by evaluating model *f* by using only a subset of features *S*. The rest of them are integrated out by using a conditional expected value formulation:$$E[f\left(X\right)|do\left(X_{S}=x_{S}\right)]$$

As seen in Figure 7, the most impactful feature is out-degree, which can be expected since a node with a high out-degree has the highest chance to activate multiple other nodes, which might result in a big initial cascade. The second and the third places could also be expected since they indicate how big the second cascade can potentially be (in the case of average neighbour degree) and how many nodes can be reached from a particular node through outgoing links (local reaching). It is also worth noting that the threshold also ranked among the top five features. This shows how important it is to assess the spreading abilities of the examined network appropriately, i.e., if we influence our friends to watch a movie (higher activation probability) or to change their political preferences (lower activation probability).

Initially, the low positions of VoteRank, clustering coefficient and PageRank were surprising to us. However, for VoteRank, the explanation might be that we were trying to predict if a particular node is influential or not, while the VoteRank algorithm is designed to produce a seed set of a particular size, and the selection of each node affects the ability of its neighbours to be selected; thus, if we select just one node, we basically select the node with the highest in-degree (and the VoteRank Shapley value is almost the same as that of in-degree).

In the case of the clustering coefficient, we can see that, in most cases, it does not matter how densely connected the seeds’ neighbourhood is. This is very interesting since, usually, we would expect that a denser neighbourhood will allow for a higher chance of cross-activation by someone from the neighbourhood if the seed fails to activate the neighbour directly.

Finally, at the second to last position, we have PageRank, which is usually considered as a measure able to select the most important users. However, if we look at the PageRank mechanism, we notice that it usually benefits nodes which have high in-degree (and, of course, important neighbours), which, as we see, is less important from the influence spreading, from an IC model point of view.

## 5. Discussion

In this paper, we presented an enhanced machine learning framework for influential node classification. We analysed currently developed approaches and proposed improvements in the areas of estimating node influence, obtaining the labels and training embedding. We also introduced new measures for defining top spreaders, which are nodes that have the most activated neighbours in a single iteration or based on the number of iterations needed to reach this peak. We examined the ability of machine learning models to generalize between different testing scenarios. Our results show that it is possible to train an ML model on one network and use it on another one. An interesting observation is that, to produce good results, it is more important to use the network from the same family than a network of similar characteristics in terms of no. of nodes/edges. This work has shown that the usage of *Smart Binning* to obtain ground-truth values for training machine learning models in key node classification tasks significantly improves the inference process in terms of higher and more stable results. Finally, we presented an analysis of feature importance from our experiments, where we observed that the ML model does not use just one feature but balances a few of them together, and that the ML model “prefers” measures that allow it to assess how many nodes could be activated in the first (out-degree), second (average neighbour degree) and all (local reaching) cascades.

Beyond all the innovations, this work exhibits strong potential for future work. For a start, it is worth to consider modern, graph-centred machine learning algorithms, such as graph neural networks, for the classification phase of our framework to measure the impact of *Smart Bins* in the most advanced node classifiers. Another possible direction for additional exploration is to further examine which features are important in terms of the generalization of ML models. Such follow-up studies could help to identify centralities that are mere noise and can be discarded. Moreover, we would like to evaluate if there are any network-similarity measures that would allow us to precisely select smaller training networks that can produce good results even for the larger networks. Such an approach would significantly reduce the overhead costs of training the ML algorithms. To take this even further, it is worth examining the effectiveness of synthetically generated networks, which may be the answer to the limited number of available real-world networks.

## Figures and Tables

**Figure 1 entropy-26-00955-f001:**
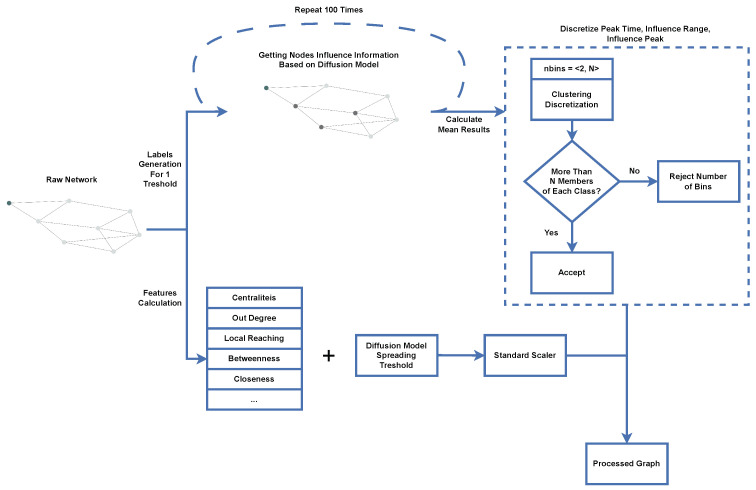
The framework for identifying influential nodes.

**Figure 2 entropy-26-00955-f002:**
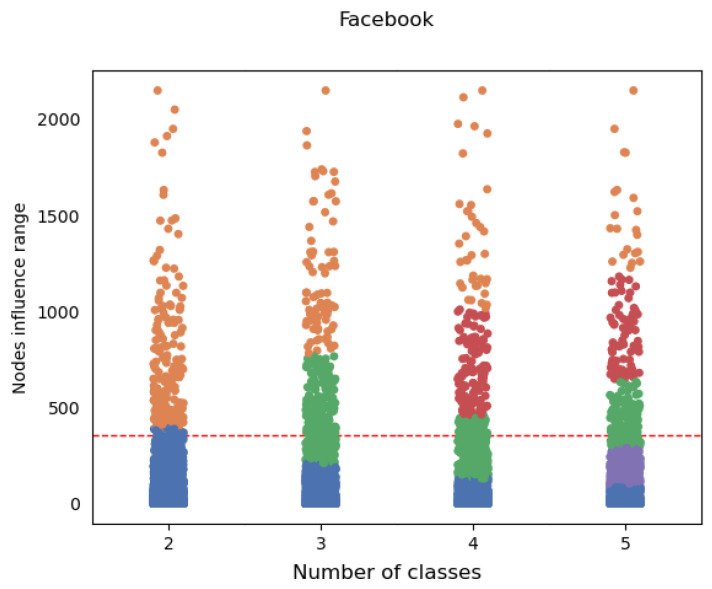
Smart bins—clustering discretization on the Facebook network. The dashed line marks the top 5% of the nodes. Nodes are colored according to the class they fit into, with the blue ones being top class (most influential nodes).

**Figure 3 entropy-26-00955-f003:**
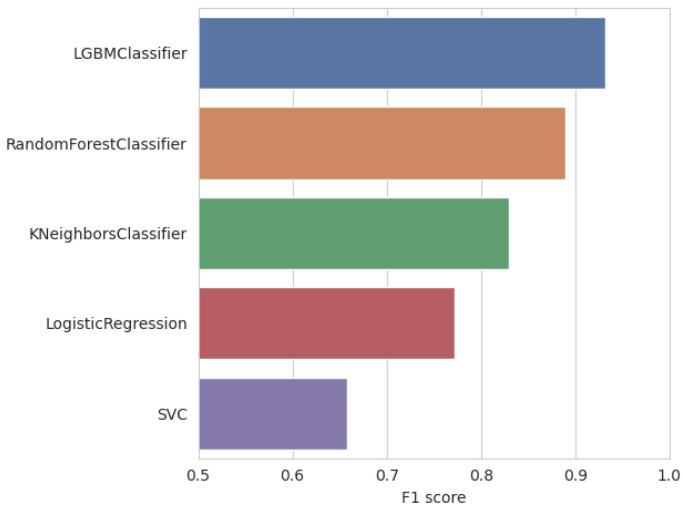
Machine learning algorithm comparison with mean values aggregated across all the experiments.

**Figure 4 entropy-26-00955-f004:**
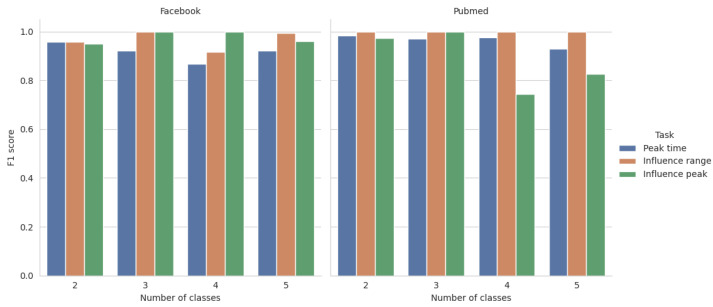
LightGBM performance in various tasks.

**Figure 5 entropy-26-00955-f005:**
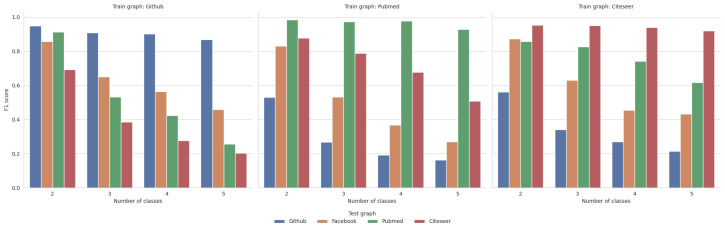
Model generalization.

**Figure 6 entropy-26-00955-f006:**
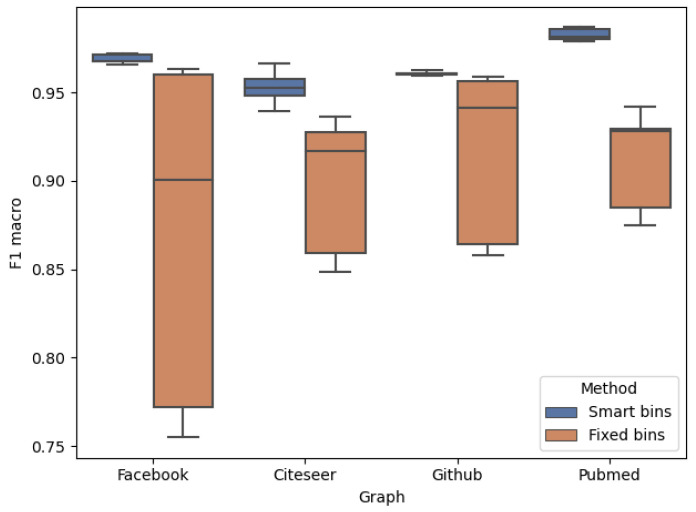
Comparison of *Smart Bins*, clustering discretization and *Fixed bins*, an arbitrary choice of top 5% nodes.

**Figure 7 entropy-26-00955-f007:**
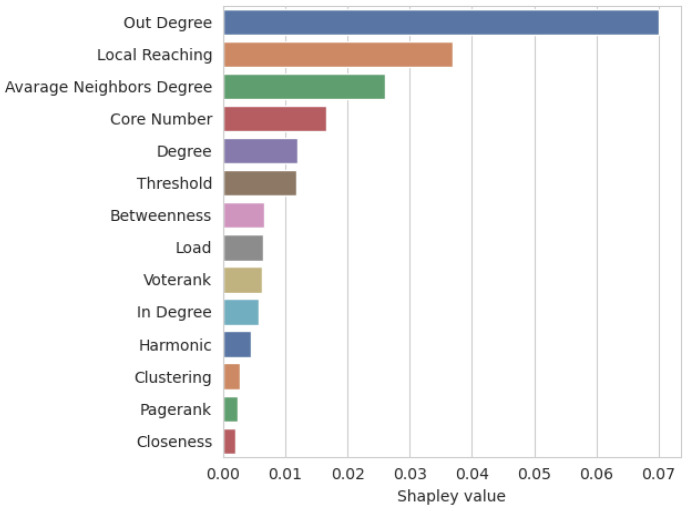
Feature importance.

**Table 1 entropy-26-00955-t001:** Basic properties of experimental networks.

Graph	Type	Nodes	Edges	Avg. Degree	Clustering Coefficient	Diameter	Transitivity
Citeseer	Cite	3327	4536	2	0.07	-	0.13
Pubmed	Cite	17,717	44,335	4	0.03	18	0.05
Facebook	Social	22,470	170,823	15	0.17	15	0.23
Github	Social	37,700	289,003	15	0.08	11	0.01

**Table 2 entropy-26-00955-t002:** The percentage of the nodes that influences range values fits into the top bin (i.e., the most influential nodes). Based on the influence range values, two, three, four and five bins were built (using KMeans). For each dataset description, please see Section 4.

Graph	2 bins	3 bins	4 bins	5 bins
Citeseer	4.170%	1.387%	0.837%	0.157%
Pubmed	3.247%	0.707%	0.247%	0.077%
Facebook	4.267%	1.687%	0.827%	0.467%
Github	10.397%	4.337%	2.477%	1.727%

**Table 3 entropy-26-00955-t003:** Centrality measures selected to describe each node.

Measure	Description
Degree	The number of connected edges to a particular node
In-degree	The number of incoming edges from nearest neighbours
Out-degree	The number of outcoming edges to nearest neighbours
Average neighbour degree	Average degree of nearest neighbours
Closeness [28]	Distance to other nodes
Betweennes [29]	Fraction of all pairs’ shortest paths that pass through the node
Local reaching [30]	It is a proportion of other nodes that are reachable from the node
VoteRank [31]	Ranking of the nodes based on voting scheme
Load [32]	The fraction of shortest paths that pass through the node
Clustering coefficient [33]	The fraction of possible edges between node neighbours
Core number [34]	It is the largest value of a maximal subgraph containing a node
Eigenvector [35]	Computes the centrality for a node based on its neighbours’ centrality
PageRank [36]	Computes a ranking of nodes based on the structure of the incoming edges
Harmonic [37]	It is the sum of the reciprocal of the shortest path distances from the node to all others

## Data Availability

The data presented in this study are available on request from the corresponding author. The data are not publicly available due to copyright reasons.

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
