# Peer review of "Identifying Key Nodes for the Influence Spread Using a Machine Learning Approach"

_entropy, 2024, doi:10.3390/e26110955_

Round 1
Reviewer 1 Report
Comments and Suggestions for Authors
In the paper entitled “Identifying Key Nodes for Influence Maximization, Machine Learning Approach”. The work presented in this paper seems good, however, I have the following major concerns:
1. The title of the paper is confusing because the authors use the well-known term "Influence Maximization," yet there is no substantive discussion of this concept in the paper. If the authors do not intend to address this term, it should be removed from the title. A simpler and more informative title would better suit the content of the paper.
2. In the abstract, the authors do not clearly specify the problem the authors are attempting to solve. It would be helpful if the authors outlined the issues in previous research and clearly stated how their work aims to address those gaps.
3. The authors claim that they introduce “alternative ways to define influential nodes by selecting them based on spreading peak and time, which have not been used yet.” Please expand on how this method is novel compared to existing approaches.
4. It is recommended to apply statistical testing to evaluate the performance of the various methods, including the proposed approach, to ensure a more rigorous analysis of results.
5. The type of dataset used (e.g., weighted, directed, or undirected) is not clearly specified in the paper. Please provide detailed information about the dataset. Additionally, it would be beneficial to include results from a synthetic dataset.
6. Why was a well-known evaluation criterion like the Kendall correlation coefficient not used? Please explain the rationale behind this decision.
7. It would be appreciated if the code for this study could be made available on open-source platforms such as CodeOcean or GitHub.

Author Response
First, we would like to thank both Reviewers for the time and effort they put into reviewing our manuscript. Your comments helped us improve our work, and we hope that you will find our answers to your comments satisfactory.
Comments 1: The title of the paper is confusing because the authors use the well-known term "Influence Maximization," yet there is no substantive discussion of this concept in the paper. If the authors do not intend to address this term, it should be removed from the title. A simpler and more informative title would better suit the content of the paper.
Response 1: We have rephrased the title to Identifying Key Nodes for the Influence Spread, Machine Learning Approach".
Comments 2: In the abstract, the authors do not clearly specify the problem the authors are attempting to solve. It would be helpful if the authors outlined the issues in previous research and clearly stated how their work aims to address those gaps.
Response 2: The main problem we are tackling in this paper is influence spread, with emphasis on identifying key nodes for the Independent Cascade Model which is a popular reference method. To qoute the abstract: „ What information about the influencers can be extracted from the network? How to precisely obtain labels required for training? Can these models generalize well? In this paper, we answer these questions by presenting an enhanced machine learning-based framework for influence spread problem.”
Comments 3: The authors claim that they introduce “alternative ways to define influential nodes by selecting them based on spreading peak and time, which have not been used yet.” Please expand on how this method is novel compared to existing approaches.
Response 3: All of the related works that we analyzed were using percentage of the activated nodes in the network as evaluation metric which corresponds to "Influence range" used in our paper. Some of them are listed below:
• Bucur, D. Top influencers can be identified universally by combining classical centralities. Scientific reports 2020, 10, 1–14.
• Zhao, G.; Jia, P.; Huang, C.; Zhou, A.; Fang, Y. A machine learning based framework for identifying influential nodes in complex networks. IEEE Access 2020, 8, 65462–65471.
• Qiao, T.; Shan, W.; Zhou, C. How to identify the most powerful node in complex networks? A novel entropy centrality approach. Entropy 2017, 19, 614
• Zhang, J.; Zhang, Q.; Wu, L.; Zhang, J. Identifying influential nodes in complex networks based on multiple local attributes and information entropy. Entropy 2022, 24, 293
• Liu, C.; Fan, C.; Zhang, Z. Finding influencers in complex networks: an effective deep reinforcement learning approach. The Computer Journal 2024, 67, 463–473.
• Zhang, J.X.; Chen, D.B.; Dong, Q.; Zhao, Z.D. Identifying a set of influential spreaders in complex networks. Scientific reports 2016, 6, 27823.
These papers do not evaluate node influence as broadly as we do – they omit key aspects such as influence peak and peak time. We believe that these two characteristics are crucial when selecting seed nodes in practical applications. For instance, consider a presidential campaign: gaining the support of a 50% of society would be ineffective if it would happen four months after the election. We have revised the section estimating node influence to clarify our reasoning behind the introduction of these metrics.
Comments 4: It is recommended to apply statistical testing to evaluate the performance of the various methods, including the proposed approach, to ensure a more rigorous analysis of results.
Response 4: Due to time constraints, we were unable to add statistical tests. However, we believe that such tests can be omitted since, as Figure 6 demonstrates, there is a significant difference between our method both in terms of the central tendency and the variance of the distribution.
Comments 5: The type of dataset used (e.g., weighted, directed, or undirected) is not clearly specified in the paper. Please provide detailed information about the dataset. Additionally, it would be beneficial to include results from a synthetic dataset.
Response 5: We have extended Table 3 to include more information about datasets. We agree that including a synthetic network would be beneficial for this kind of research. Unfortunately, doing so properly is a task for a completely separate paper. We would need to generate a few networks of various sizes (e.g. 100, 1000, 1000 nodes), using various network models (e.g. ErdÅ‘s–Rényi, Watts–Strogatz, Barabási–Albert, etc.) and various parameters for each model. This would result in at least 300 new networks. For each network, we would have to calculate the ground truth (i.e. calculate the
maximum spread for each node), train models, etc. The ground truth calculation alone would take at least 2 months. That is why in the oryginal paper we have included synthetic network evaluation as a future work. Please, see the last sentence of the Discussion section.
Comments 6: Why was a well-known evaluation criterion like the Kendall correlation coefficient not used? Please explain the rationale behind this decision.
Response 6: In our evaluation, the F1 measure is considered more reliable for the presented analysis. It is a commonly used metric in various machine learning applications and provides a robust basis for drawing reliable conclusions. There are few crucial differences between the F1 score and Kendall correlation that led us to choose the F1 measure for evaluating our results:
• The F1 score is specifically designed for classification tasks, which aligns with the nature of the problem we are addressing, whereas Kendall’s tau is more suited for ranking or correlation tasks.
• The F1 score is particularly useful when there is an imbalance in class distribution, as it balances precision and recall, which is important in our context.
• The Kendall correlation coefficient is commonly used in statistical analyses, especially with ordinal data, to assess the similarity between rankings.
Comments 7: It would be appreciated if the code for this study could be made available on open-source platforms such as CodeOcean or GitHub.
Response 7: The code is now available in the repository linked below. It has been added to the manuscript at the end of introduction.
https://github.com/mateuszStolarski/identifying-key-nodes-influence-spread-ml
Again, we would like to thank the Reviewers for their precious and detailed comments. By incorporating them into the manuscript, we believe its quality has improved and hope that the Reviewers will share the same opinion.

Reviewer 2 Report
Comments and Suggestions for Authors
See up loaded file.

Author Response
First, we would like to thank both Reviewers for the time and effort they put into reviewing our manuscript. Your comments helped us improve our work, and we hope that you will find our answers to your comments satisfactory.
The goal is to have a machine learning, ML, process identify influential nodes in large networks. This is a hard problem and an important one. (I looked forward to reviewing it.). Their presentation should:
1) explain what constitutes an influential node,
2) describe the training routine, in particular the nature of feedback,
3) the accuracy of results.
For step (1) they adopt a variant, IC, of the SIR model this is often used in epidemiology research
to describe the spread of infection. It is a good choice. They display the 3 differential equations usually associated with this SIR model (line 120), but
Comment
• never really use them to obtain “ground truth values for a given node” (line 138) .
Replay
That is true. We have introduced the SIR model as it is most often used in related works, and explain why we have decided to use IC instead (since related papers use SIR with recovery rate 1 effectively changing it to the IC model). Then we used IC model to obtain Ground-truth. We prefer to leave the section about SIR in the paper as it allows the reader to put our work in perspective to previous works. A the same time, we want to emphasize that our framework is not restricted to the IC model - e.g. SIR could have been used there just as well.
Comment
• Was this for every node in thee large networks?
Replay
Yes, we have calculated Ground-truth for every node in each network we evaluated.
Comment
• What were the measured characteristics? I presume they were “Influence range”, “influence peak” and “peak time” (lines 151-157).
Replay
Yes, we have measured all those characteristics. Influence range is a characteristic that can be found in every paper that explores this problem, but we wanted to bring attention to the other aspects of measuring influence, and that is why we introduced influence peak and peak time.
Comment
• How were these used to “generate labels” (line 128) and what were the labels?
Replay
During IC model simulations we were collecting raw (numerical) information about nodes which are:
– Influence range – predicting the total range, i.e., the number of influenced nodes in the network. This is the main task – widely recognized in the relevant literature.
– Influence peak – predicting the maximal number of nodes activated in one iteration. Crucial characteristic from the practical perspective (e.g. total number of people infected by a virus at once – allows to approximate the woarkload for medical services.)
– Peak time – predicting the number of iterations required for the process to reach the Influence peak. Important value for real-life use cases.
When we had finished simulations we were able to aggregate those values and discretize them using smart-bins to obtain node labels.
Comment
In Section 3.2 “Obtaining the labels”, the authors claim that “Smart bins” is their main contribution (line 176) but it is unclear to me what fills a smart bin. Based on Figure 1 (p. 3) (which I find most obscure) and the flow chart decision at the right I assume a “smart bin” corresponds to a cluster of network nodes. It appears that a number of “smart bins” are chosen and then the Kmeans algorithm (equation line 185 should read “min” rather than “max”) is used to populate the bin (class?) as a cluster. But this is belied by Figure 2 where presumably each color corresponds to a bin (class?) and each dot a node? But if so then the “top 5% of nodes” are in one bin with limited “influence range””? I am mystified. My own experience with clusters in citation graphs is that there tend to be many small relatively tight clusters, so limiting the process of 5 smart bins or less doesn’t make sense. Yet, understanding the nature of “Smart bins” seems to be crucial to understanding step (2) above and the paper.
Replay
For the sake of this answer, let’s assume the node’s influence is the total number of nodes it activated during the spreading process, but the algorithm works the same for influence peak or peak time.
The smart binning process, as described in the paper is as follows:
1. We calculate influence for each node.
2. We discretize obtained values by using KMeans clustering methods.
Key factor here is the fact, that we cluster nodes’ influence, not the nodes themselves (as in graph clustering). In such cases, there is no phenomenon of many tight clusters; quite the opposite. Usually there is a small cluster of top-spreaders, many medium-spreaders and a few no-spreaders etc. The K parameter in KMeans allows as too choose the granularity of this discretization, e.g. we can choose to only identify top-spreaders and others (K=2). We extended section 3.2 to explain this process in more detail.
What is important in our research – this approach is vastly superior to the one used in existing papers. There, the authors arbitrarly choose top (e.g.) 5% of the nodes assuming that those nodes are a distinct group, with more spreading potential then others. In our paper (most visible in figures 2 and 6) we prove their assumptions wrong.
Comment
Table 2 (Section 3.3, line 215) is a complete list of centrality measures, which are often assumed to
correspond to “influence”; but only the first 4 and “clustering coefficient” are local (I.e.easily computed for each node) and useful for ML feed back. Were the others use for ML feed back or were they just used for final evaluation?
Replay
As node embeddings, we used all presented features across all our experiments. It was important for us to maintain a consistent set of features for the final evaluation, which was based on feature importance. In our case, generating these features was not computationally expensive compared to generating the ground truth. However, their inclusion led us to the conclusion that some centrality measures, which are intuitively useful in the influence maximization problem, such as Pagerank, are not practical to calculate in real-world solutions.
Comment
Figure 3 (Section 4.1, line 240) Since the F1 score is a measure of precision and recall, how are the “mean values aggregate across all the experiments” how are these determined for the figure? That a process could have an F1 score of 1.0 for “influence range” seems somewhat incredible.
Replay
We simply aggregated F1 scores achieved by an ML model of given type and then calculated a mean value to achieve a general comparison of chosen ML algorithms. An F1 score of 1.0 is indeed hard to achieve, but theoretically possible - this is why it is visible on the x-axis (but not reached by out models).
Comment
The section 4.4 “Features Importance” (line 292) is quite interesting, if only there was a clear explanation of how they derived it. I suspect it is true.
Replay
We agree that the results of the features’ Importance evaluation are interesting. To derive/assess, features importance
we have used Shapley values. Using Shapley values, we can compute each feature’s contribution to the final prediction because they represent how much the presence or absence of a feature changes the predicted output of the ML model for a particular input. Shapley values are obtained by evaluating model f using only a set subset of features S. The rest of them are integrated out by using a conditional expected value formulation:
E[f(X)|do(XS = xS)]
Below we include more detailed explanation of Shapley values based on the paper "Shapley, L. S. (1953). A value for n-person games. Contribution to the Theory of Games, 2." (which is referenced in the manuscript).
Shapley values come from game theory and help us figure out how much each player (or factor) contributes to a group’s overall success. To make this easier to understand, let’s imagine the following: You and three friends are making a pizza. Each of you brings one ingredient: dough, sauce, cheese, and toppings. The pizza won’t be complete unless everyone contributes something. But maybe one ingredient, like the dough, is more important because you can’t make a pizza without it at all. So, how do we decide how important each person’s contribution is?
That’s what Shapley values help to calculate. They tell us how much each person contributes to the pizza by considering all possible ways to combine the ingredients. It doesn’t just look at one way of making the pizza, but all possible ways, and then averages out how much each ingredient helped. In more technical terms, if you have a system or model with many different inputs (like a machine learning model), Shapley values tell you how much each input (or feature) contributes to the final prediction or outcome.
Key features
• Fairness: Shapley values provide a fair way to allocate contributions among the different players or features.
• All Combinations Matter: The value considers every possible combination of inputs to see how each input contributes, whether it’s used alone or with others.
• Average Contribution: Shapley values are the average contribution of a feature across all possible scenarios in which that feature appears.
How we interpret them: If a feature (ingredient) has a high Shapley value, it means it is important and has a big impact on the outcome. If a feature has a low or negative Shapley value, it means it contributes little or even decreases the outcome.
In summary, Shapley values help us fairly assign credit to different factors in a system, like figuring out how important each friend’s ingredient was for making the pizza. They are widely used in machine learning to explain how different variables affect the model’s predictions.
The English is of high quality and the only typo I found is “woarkload” (line 155).
Thank you for this comment, we have corrected the typo and combed the paper one more time using Grammarly and Writefull to check for typos.
Again, we would like to thank the Reviewers for their precious and detailed comments. By incorporating them into the manuscript, we believe its quality has improved and hope that the Reviewers will share the same opinion.

Round 2
Reviewer 1 Report
Comments and Suggestions for Authors
The authors have incorporated the required changes, so the paper can be accepted.
Author Response
Once again, we thank both Reviewers for the time and effort they put into reviewing our manuscript and replays for the first round of reviews. Below, include the replays for Reviewers’ comments from the second round of reviews.
Comments 1: The authors have incorporated the required changes, so the paper can be accepted.
Reply 1: Thank you

Reviewer 2 Report
Comments and Suggestions for Authors
Please check the attachment.

Author Response
Once again, we thank both Reviewers for the time and effort they put into reviewing our manuscript and replays for the first round of reviews. Below, include the replays for Reviewers’ comments from the second round of reviews.
Comments 1: I am uncertain how to respond to this new revision. The authors have added verbiage, such as “using the Independent Cascades Model”, about which I can find no information — and why wasn’t this mentioned in the first submission?
Reply 1: The Independent Cascade Model (IC) is an example of diffusion models that estimate the influence a node or a set of nodes can exert. The main assumption of the IC model is that each node has only one opportunity during the simulation to spread its signal further in the network. This provides simpler assumptions for generating labels in our work, as there is only one parameter (the spreading threshold), in contrast to epidemic models such as SIR. The Independent Cascades Model was mentioned in section 3.1 in all versions of the manuscript (from line 124 in the first version and from line 127 in the second version). We have further modified this section.
Comments 2: I still don’t understand what a “smart bin” is, even though this is their “main contribution” (line 189). I see no effort in this resubmission to explain what a smart bin is, even though it was my primary criticism of their submission. (I still think these bins might contain network nodes, but I’m not sure.)
Reply 2: Bins do not contain nodes. Bins are calculated based on nodes influence range values and are a way to split some space into ranges. For example, the first bin can have a range from 1 to 10, the second from 11 to 20, etc. Based on each node’s influence range value, we can check to which bin it matches and, using that, assign the proper label (class) to that node. We have added a new paragraph to section 3.2 that better demonstrates what the bin is and the difference between bin, class and label. The “smart bin” is an approach of building bins automatically based on data and not some arbitrary choice (like in the case of related papers).
Comments 3: In the original version, in Section 3.2 “Obtaining the labels” the only further mention of “labels” was “quantile discretization to obtain a set of K labels” (line 170). The revision adds 9 new lines (161-169), but not clarification as to what labels are to be obtained nor their actual use, even though this was also a question in my original review.
Reply 3: Labels and features are the most fundamental concepts in machine learning. For each model, one needs a set of features (the question) as an input and the labels (the answer) as the baseline output. We have added a new paragraph to section 3.2 that better demonstrates what the bin is and how it is "translated" into class and label.
Comments 4: I try to learn from every paper I referee. I had hoped to learn how machine learning, ML, can identify central nodes in large networks, and issue I have previously struggled with. But, I simply can’t comprehend how this process works, so I doubt if any others who are not privy to the buzz words and phrases will either.
Reply 4: We hope that an additional explanation of ML terms makes our process understandable. Below is a short description of our framework
Following explanation is only a simplified summary; the detailed explanation can be found in the paper.
1. We use the Independent Cascade model to obtain the influence range value for each node (sec 3.1)
2. Here, we could have created a regression ML model and learn to predict this value (and, by extension, identify the nodes with the most spreading potential), but instead we opt for a classification model (sec 3.2, paragraph 1 explains the motivation). To do that we need to obtain the labels by discretizing the influence range values.
3. We introduce our approach to this problem (sec 3.2, paragraphs 4+) – utilizing unsupervised learning techniques (KMeans algorithm as an example) to discretize the influence range values in a more flexible way.
4. At this stage we have all it takes to create a ML classifier: ground truth (i.e. labels) – discretized (by smart bins, sec 3.2) influence range values and node’s features (sec 3.3).
5. Finally (in sec 4.3) we present the impact made by smart binning on the performance of the trained models. By comparing our approach with those found in other papers.
Comments 5: As a final example consider Table 1, p.5, “Percentage share of the top class in all network[s]”. Nothing in the text explains its meaning. What is the “top class”? The number of bins appears to make a huge difference. Why? Why are only 0.15 (of the nodes?) in Citeseer in the top class if there are v5 bins? It cries for some explanation, but the reader is presumed to know.
Reply 5: Top class means the class containing top influencers (i.e. nodes with the highest spread potential). We rephrased the Table 1 caption to be more clear and in line with the new content in section 3.2.
The 0.157% of nodes as in a top class (i.e. nodes that influence range value fit into the top bin) in a Citeseer is a result of forcing the algorithm to create 5 bins. The algorithm iteratively groups a bigger set of data points into smaller subsets in such a way that points within a single cluster are more similar to each other than to representatives of other clusters. In this case, the algorithm found that with the division into 5 bins, the influence range values of 0.157% of nodes are different than the influence range values of the rest of the nodes.
Comments 6: This is a frustrating paper. It promises a lot, but never really delivers. I cannot recommend its acceptance.
Reply 6: Thank you for taking the time to review our work and for your comments. We appreciate your candor, and we hope that changes to the paper and our responses have clarified all issues.
